# SARS-CoV-2-Induced TSLP Is Associated with Duration of Hospital Stay in COVID-19 Patients

**DOI:** 10.3390/v15020556

**Published:** 2023-02-17

**Authors:** Luke Gerla, Subhabrata Moitra, Desmond Pink, Natasha Govindasamy, Marc Duchesne, Eileen Reklow, Angela Hillaby, Amy May, John D. Lewis, Lyle Melenka, Tom C. Hobman, Irvin Mayers, Paige Lacy

**Affiliations:** 1Alberta Respiratory Centre (ARC) Research, Division of Pulmonary Medicine, Department of Medicine, University of Alberta, Edmonton, AB T6G 2R3, Canada; 2Nanostics Inc., Edmonton, AB T6G 2E9, Canada; 3Department of Oncology, University of Alberta, Edmonton, AB T6G 2H6, Canada; 4Entos Pharmaceuticals, Edmonton, AB T6G 3Q5, Canada; 5Department of Cell Biology, University of Alberta, Edmonton, AB T6G 2H6, Canada; 6Synergy Respiratory Care, Sherwood Park, AB T6G 2E9, Canada

**Keywords:** coronavirus, cytokine, TSLP, IL-15, CXCL10, human bronchial epithelial cells

## Abstract

Thymic stromal lymphopoietin (TSLP) is an epithelium-derived pro-inflammatory cytokine involved in lung inflammatory responses. Previous studies show conflicting observations in blood TSLP in COVID-19, while none report SARS-CoV-2 inducing TSLP expression in bronchial epithelial cells. Our objective in this study was to determine whether TSLP levels increase in COVID-19 patients and if SARS-CoV-2 induces TSLP expression in bronchial epithelial cells. Plasma cytokine levels were measured in patients hospitalized with confirmed COVID-19 and age- and sex-matched healthy controls. Demographic and clinical information from COVID-19 patients was collected. We determined associations between plasma TSLP and clinical parameters using Poisson regression. Cultured human nasal (HNEpC) and bronchial epithelial cells (NHBEs), Caco-2 cells, and patient-derived bronchial epithelial cells (HBECs) obtained from elective bronchoscopy were infected in vitro with SARS-CoV-2, and secretion as well as intracellular expression of TSLP was detected by immunofluorescence. Increased TSLP levels were detected in the plasma of hospitalized COVID-19 patients (603.4 ± 75.4 vs 997.6 ± 241.4 fg/mL, mean ± SEM), the levels of which correlated with duration of stay in hospital (β: 0.11; 95% confidence interval (CI): 0.01–0.21). In cultured NHBE and HBECs but not HNEpCs or Caco-2 cells, TSLP levels were significantly elevated after 24 h post-infection with SARS-CoV-2 (*p* < 0.001) in a dose-dependent manner. Plasma TSLP in COVID-19 patients significantly correlated with duration of hospitalization, while SARS-CoV-2 induced TSLP secretion from bronchial epithelial cells in vitro. Based on our findings, TSLP may be considered an important therapeutic target for COVID-19 treatment.

## 1. Introduction

Thymic stromal lymphopoietin (TSLP), an epithelium-derived pleiotropic cytokine, is considered a master regulator of the early innate immune response to a range of airborne environmental factors and pathogens. The release of TSLP from the airway epithelium stimulates innate lymphoid cells, T cells, dendritic cells, and mast cells to augment their effector functions and promote the amplification of innate and adaptive immune responses in infected tissues [1,2,3]. In patients with coronavirus disease (COVID-19), it is believed that the innate immune response to severe acute respiratory syndrome coronavirus 2 (SARS-CoV-2) can overwhelm the body in a proportion of infected individuals and result in a catastrophic reaction to the virus. This excessive production of pro-inflammatory cytokines in response to SARS-CoV-2 is well documented and can precipitate an acute respiratory distress syndrome (ARDS)-like condition requiring intensive care [4,5,6,7]. 

A major cell type in the lung innate immune system is the airway epithelial cell, which has the capacity to produce a wide range of alarmins and pro-inflammatory cytokines in response to many triggers including respiratory viruses that can rapidly overwhelm immune responses. TSLP, interleukin (IL)-25, and IL-33 are three alarmin-type cytokines that are initially generated by epithelial cells in response to infection by RNA viruses including coronaviruses [8,9,10,11,12]. It should be mentioned that these studies have not directly measured TSLP production from cultured airway epithelial cells infected specifically with SARS-CoV-1 or SARS-CoV-2; most studies to date have focused on the related RNA virus, respiratory syncytial virus, which also induces TSLP production. While some studies suggest that little to no TSLP is detectable in epithelial cell supernatants or the plasma of infected patients during SARS-CoV-2 infection [4,13], others showed an elevation of TSLP in the circulatory system of COVID-19 patients [5,14,15]. Moreover, no reports have shown whether SARS-CoV-2 infection stimulates TSLP expression in epithelial cells. 

In this study, we investigated the production of TSLP during SARS-CoV-2 infection in hospitalized COVID-19 patients and its association with other factors including duration of hospital stay. The effects of SARS-CoV-2 on TSLP production was also determined in cultured human nasal, intestinal, and bronchial epithelial cells.

## 2. Materials and Methods

### 2.1. Study Population for Collection of Plasma Samples

Patients testing positive for SARS-CoV-2 (*n* = 11, confirmed COVID-19-positive by polymerase chain reaction testing for spike protein (SP), provided by Alberta Health Services) and experiencing symptoms requiring hospitalization were admitted to the non-intensive care unit (non-ICU) of the University of Alberta Hospital in Edmonton, Alberta, Canada in the period 18 October–25 November 2020. Clinical parameters upon hospitalization such as heart rate, respiratory rate, body temperature, oxygen saturation in blood (SpO_2_), comorbidities, and the interval between the date of the COVID-19-positive test result and the date of hospitalization were assessed from electronic medical records on the first day of hospitalization. 

### 2.2. Inclusion and Exclusion Criteria

Inclusion criteria were that (1) participants provided informed consent prior to any study specific procedures, (2) they were 18–80 years old, and (3) that they were admitted to hospital for COVID-19 symptoms. Exclusion criteria were any participants (1) receiving an investigational drug product within five half-lives of the drug, (2) receiving inhaled corticosteroids prior to hospital admission, (3) receiving biological agents within the preceding six months, or (4) unable to provide informed consent. 

### 2.3. Collection of Blood Samples

Venous blood samples were collected from nonvaccinated COVID-19-positive patients (*n* = 11) on the first date of hospitalization into BD Vacutainer^®^ EDTA tubes (BD Canada, Mississauga, ON, Canada) and centrifuged at 1000× *g* for 15 min at 4 °C. For comparison, we recruited 10 age- and sex-matched healthy control participants without a history of COVID-19 infection for plasma collection. Plasma supernatants were collected, aliquoted, and snap-frozen on dry ice and ethanol for storage at −80 °C before sample analysis. Plasma samples from COVID-positive patients were also collected from the Canadian BioSample Repository, which is a biobank facility located at the University of Alberta campus (https://biobanking.org/canreg/view/407, accessed on 10 March 2021). 

### 2.4. Normal Human Bronchial Epithelial (NHBE) Cell Cultures

Human bronchial epithelial cells (normal human bronchial epithelial cells (NHBE)) and the intestinal cell line Caco-2 from the American Tissue Culture Collection (ATCC) as well as nasal epithelial cells (HNEpC) from PromoCell (Heidelberg, Germany) were cultured under sterile conditions in bronchial epithelial growth media (BEGM) plus supplements (bovine pituitary extract, insulin, hydrocortisone, retinoic acid, transferrin, triiodothyronine, epinephrine, human epidermal growth factor, penicillin, streptomycin) at 37 °C with 5% CO_2_ in a humidified incubator. Cells were cultured on ethanol-washed glass coverslips until they reached ~80% confluency. Attached cells were then subjected to mock infection (vehicle-treated with BEGM media) or SARS-CoV-2 virus infection (SARS-CoV-2/Canada/VIDO-01/2020, originally cultured by VIDO-Intervac, University of Saskatchewan, SK from a clinical specimen originating from Sunnybrook Health Sciences Centre, Toronto, ON and propagated at the National Microbiology Laboratory, Winnipeg, MB; MOI 1) for 24 h in a biosafety level 3 containment facility in a humidified incubator at 37 °C and 5% CO_2_. Infection was terminated by the addition of 4% buffered paraformaldehyde, and coverslips were then processed for immunolabeling. 

### 2.5. Human Bronchial Epithelial Cells (HBECs) from Bronchial Brushings Derived from Non-Infected Participants

To test TSLP release from freshly isolated airway epithelial cells, we obtained human bronchial epithelial cells (HBECs) from bronchial brushings from six noninfected individuals undergoing elective bronchoscopies for unidentified lung masses at the University of Alberta Hospital Endoscopy unit. Patients with suspected or confirmed SARS-CoV-2 infection were excluded from sample collection. Bronchial brushings were collected using cytology brushes from the surface of apparently healthy right or left main stem bronchus that was distant from any apparent lung mass and were placed in BEGM transport medium to be subsequently analyzed. Samples were then vortexed, and brushes were removed from media. HBECs were centrifuged at 300× *g* for 5 min at 4 °C, and the pellet was resuspended in BEGM before being transferred to coverslips or flasks and grown to ~80% confluency prior to infection as described above for NHBE cells.

### 2.6. Assays for Cytokines

Frozen plasma samples were thawed and subjected to analysis using Meso Scale Discovery Multiplex Assay kits (MSD, Rockville, MD, USA) following manufacturer’s protocols. Specific kits used in this report were an ultra-sensitive S-Plex human TSLP kit with a lower limit of detection (LLOD) of 9.1 fg/mL (lower limits of quantification (LLOQ) 34 fg/mL), with an approximately 1000-fold higher sensitivity than other commercially available assay kits for human TSLP. V-Plex human Cytokine Panel 1 biomarker multiplex kits were also used for IL-1α, IL-5, IL-7, IL-12/IL-23p40, IL-15, IL-16, IL-17, vascular endothelial growth factor A (VEGF-A), tumor necrosis factor b (TNF-β)/lymphotoxin α (LTα), CCL2/monocyte chemoattractant protein 1 (MCP-1), CCL3/macrophage inhibitory protein-1α (MIP-1α), CCL4/MIP-1β, CCL11/eotaxin, CCL13/MCP-4, CCL17/thymus and activation-regulated chemokine (TARC), CCL22/monocyte-derived chemokine (MDC), CCL26/eotaxin-3, and CXCL10/interferon-γ-induced protein 10 (IP-10), with an LLOD of 0.04–9.33 pg/mL.

### 2.7. Immunofluorescence for TSLP

Coverslips with attached NHBEs or HBECs were fixed in 4% buffered paraformaldehyde (pH 7.4) at room temperature for 20 min, washed with phosphate-buffered saline (PBS), and permeabilized with 0.1% Triton X-100 in PBS, pH 7.4 for 10 min. After washing and blocking in PBS with 2% bovine serum albumin and 5% goat serum, cells were labeled with primary rabbit anti-TSLP antibody (10 μg/mL, Millipore Sigma, Mississauga, ON, Canada) and in some cases double-labeled with diluted human sera obtained from volunteers who had been vaccinated against SARS-CoV-2 spike protein (SP) (Hobman lab). Primary antibodies were detected using anti-rabbit IgG conjugated to Alexa 647 for anti-TSLP (Jackson ImmunoResearch, West Grove, PA, USA) and anti-human IgG conjugated to Alexa 488 for anti-SP (Invitrogen, ThermoFisher Scientific, Waltham, MA, USA) respectively. Counterstaining was carried out with rhodamine- or Alexa Fluor 647-conjugated phalloidin stain for actin cytoskeleton and Hoechst stain for nuclei (Invitrogen, ThermoFisher Scientific, Waltham, MA, USA). The specificity of immunostaining was verified via isotype controls. All images were collected on an Olympus IX81 epifluorescence microscope and analyzed using cellSens software (Olympus Canada, Mississauga, ON, Canada), and Volocity image analysis software (version 6.3, Puslinch, ON, Canada) was used to determine fluorescence intensity. Cell outlines or regions of interest (ROI) were automatically defined using phalloidin (cytoskeleton stain) and the mean fluorescence intensity was calculated using the sum of all pixel values within each ROI (Appendix A). 

### 2.8. Statistical Analysis

Demographic, clinical, and cytokine data were described as mean (standard error of mean (SEM)), median (interquartile range (IQR)), or frequency (%) for continuous, ordinal, and categorical variables, respectively. To compare the levels of cytokines between the infected (COVID-19 patients) group and uninfected controls, we used a one-way analysis of covariance (ANCOVA) for each cytokine adjusting for sex and age. Spearman’s correlations were computed for the association between TSLP and all other cytokines with Šidák’s correction. As data regarding clinical factors were not available for the control group, we restricted the additional analyses to the infected group only. To find the association between TSLP levels and period of hospitalization (days), we used a generalized linear (Poisson) regression model. Based on prior evidence, age, sex, heart rate, respiratory rate, body temperature, and SpO_2_ on admission, number of comorbidities, and the interval between the date of the COVID-19-positive test result and the date of hospitalization were considered as potential confounders. The goodness of fit of the models was tested by Akaike’s information criteria (AIC) [16], and potential collinearity among the covariates was tested using the variance inflation factor (VIF). In addition to that, we also performed some mediation analyses. First, we identified three potential cytokines that had a higher correlation with TSLP (IL-17, CCL2/MCP-1, and CCL11/eotaxin) from the correlation matrix (without applying Šidák’s correction). Then, we performed an exploratory mediation analysis to test the influence of IL-17, CCL2/MCP-1, and CCL11/eotaxin on the association between plasma TSLP level and period of hospitalization. 

For cellular data, we presented intracellular TSLP expression as median (IQR) and analyzed the difference in TSLP release between mock- and SARS-CoV-2-infected NHBEs and HBECs by Mann–Whitney test. For the subgroup analysis including isotype controls, mock- and SARS-CoV-2-infected cells, we used the Kruskal–Wallis test with Dunn’s post hoc test for multiple comparisons. All analyses were performed in STATA v17.0 (Stata Corp, College Station, TX, USA) and violin plots were constructed in GraphPad Prism Version 9.0 (GraphPad Software, San Diego, CA, USA). A *p*-value < 0.05 was considered as statistically significant.

## 3. Results

### 3.1. Demographic and Clinical Profiles

The majority of healthy controls and hospitalized COVID-19 patients recruited for this study were males (6/10, and 8/11, respectively) and were of a similar age (57.7 ± 9.8 vs. 60.1 ± 11.9 years, respectively, mean ± SEM) (Figure 1). COVID-19 patients had multiple comorbidities (median: 3; IQR: 2–7), and the median (IQR) duration of hospitalization was 5 (3–14) days.

### 3.2. Cytokine Profile

To determine differences, if any, in cytokine levels between healthy and hospitalized COVID-19 participants, we used customized cytokine detection panels to measure cytokines in plasma samples. We observed that plasma TSLP levels increased from 603.4 ± 75.4 fg/mL in healthy participants (values for plasma samples shown as mean ± SEM) to 997.6 ± 241.4 fg/mL in hospitalized COVID-19 patients (Figure 1). Although there was a 65% increase in TSLP in the COVID-19 group than the healthy participants, this difference did not achieve the threshold for statistical significance in multivariable analysis (*p* = 0.25). We also observed a > 2-fold increase in IL-15 in COVID-19 patients (2.3 ± 0.1 for controls vs. 5.7 ± 0.8 pg/mL for COVID-19 patients) and a 10-fold increase in CXCL10/IP-10 over controls (401.5 ± 75.5 pg/mL controls vs. 4257.0 ± 1383.1 pg/mL COVID-19 patients, *p* < 0.001). In contrast, TNF-β/LTα levels were lower in COVID-19 patients (0.16 ± 0.04 pg/mL) than controls (0.39 ± 0.06 pg/mL, *p* < 0.001). A similar decrease in COVID-19 patients was observed for CCL13/MCP-4 (89.1 ± 9.0 vs. 155.9 ± 21.5 pg/mL, *p* = 0.01) and CCL22/MDC (523.9 ± 55.0 vs. 914.9 ± 109.4 pg/mL, *p* = 0.007). All other cytokines measured in this study did not differ significantly between the two groups. We observed a significant positive correlation between TSLP and IL-17 (*ρ*: 0.53; *p* = 0.02), CCL2/MCP-1 (*ρ*: 0.53; *p* = 0.02), and CCL11/eotaxin (*ρ*: 0.67; *p* = 0.002) but not with other measured cytokines. However, these associations did not remain significant after corrections for multiple testing (all *p*-values > 0.05) (Figure 2).

### 3.3. Association between TSLP and Hospitalization

We observed that higher plasma TSLP levels at the time of hospitalization were associated with an increased period of hospitalization (regression coefficient (β): 0.11; 95% confidence interval (CI): 0.01 to 0.21 per 100 fg/mL increase in plasma TSLP level, *p* = 0.03) (Figure 3). We did not observe any significant collinearity among the tested covariates (VIF < 1.8). In the exploratory mediation analyses, IL-17, CCL2/MCP-1, or CCL11/eotaxin showed no significant effects on the relationship between higher plasma TSLP levels and longer periods of hospitalization (data not shown).

### 3.4. Expression and Secretion of TSLP in Cultured AIRWAY Epithelial Cells during SARS-CoV-2 Infection

As we found some association between plasma TSLP levels and SARS-CoV-2 infection, we next investigated whether SARS-CoV-2 virus infection could induce TSLP expression using in vitro cultured bronchial epithelial cells. In cultured NHBEs, we found that compared to mock-infected NHBEs, TSLP levels increased significantly in SARS-CoV-2-infected cells after 24 h (MOI 1) (medians 877 vs. 928, respectively, *p* < 0.0001) (Figure 4A,B). This was supported by an increase in supernatant levels of TSLP following 24 h SARS-CoV-2 infection (Appendix A). Furthermore, cells that expressed SP immunofluorescence (SP^+^) indicating infection by SARS-CoV-2 virus showed a significant increase in intracellular TSLP compared to mock-infected and SP^-^ SARS-CoV-2-infected cells (*p* < 0.0001, *p* = 0.005, respectively, Figure 4C). A dose–response curve with MOI 0.1, 1, and 3 of SARS-CoV-2 showed increased TSLP expression in correlation with increased viral titer (Appendix A). An MOI of 1 was considered to be the optimal viral titer for epithelial cell infection, as MOI 0.1 showed insignificant changes in TSLP levels, while MOI 3 induced changes in cell morphology.

Similar results were obtained with HBECs. Compared to mock-infected cells, there was a significant increase in TSLP immunofluorescence in SARS-CoV-2-infected cells (medians 1326 vs. 1539, respectively, *p* < 0.0001) (Figure 5A,B). HBECs exhibited a widely distributed amplitude of TSLP immunofluorescence compared to NHBEs. No significant difference was observed in SP^+^ cells from SARS-CoV-2-infected cells compared with SP^-^ cells from within the same population (Figure 5C). However, similar to NHBEs, HBECs showed elevated TSLP immunofluorescence in SP^+^ cells over that of mock-infected cells (Figure 4C).

Additionally, we determined if other epithelial cell types that are SARS-CoV-2-compliant [17,18,19,20] express TSLP in response to infection. Cultures of primary human nasal epithelial cells (HNEpC) and Caco-2 cells were infected with increasing MOI of SARS-CoV-2 and assessed for intracellular TSLP expression and TSLP released in supernatants at 24 h. While HNEpC expressed intracellular TSLP in basal conditions, Caco-2 cells exhibited negligible TSLP immunofluorescence (Appendix A). No significant changes in supernatant levels of TSLP were observed in SARS-CoV-2-infected cells compared to mock-infected HNEpC and Caco-2 cells, while NHBE cells showed a dose-dependent increase in TSLP (Appendix A).

## 4. Discussion

Our findings indicate that plasma levels of TSLP, IL-15, and CXCL10/IP-10 were elevated in COVID-19 patients over that of normal uninfected controls. Although the difference in TSLP levels did not achieve a statistically significant level, it has to be noted that a clinically significant observation may not always satisfy statistical criteria for significance. Our analysis also suggests, for the first time, that plasma levels of TSLP at the time of hospitalization could plausibly indicate the period of hospitalization in COVID-19 patients. In parallel with this, we confirmed that SARS-CoV-2 infection induces TSLP production in cultured bronchial epithelial cells as a likely source of plasma TSLP in infected patients.

The function of TSLP appears to be that of a master cytokine that has the potential to initiate an inflammatory cascade during a virus infection, leading to the cytokine storm observed during SARS-CoV-2 infection. However, whether SARS-CoV-2 infection initiates an increase in TSLP production has been contradictory [4,5,13,14,15]. While our findings of increased plasma TSLP in COVID-19 patients are in agreement with several recent reports suggesting elevated TSLP levels in plasma from moderate and severe COVID-19 patients compared with healthy controls and influenza A (H1N1)-infected individuals [5,14,15], others reported no increase in plasma TSLP in COVID-19 patients [13] or cell culture supernatants upon infection of epithelial cells by SARS-CoV-2 [4]. One reason for this may be that some of these studies relied on less sensitive multiplex assays (detection in the pg/mL range) that missed TSLP expression. Our studies utilized a more sensitive and specific assay for TSLP measuring in the femtogram per milliliter range, which is 1000-fold more sensitive than most multiplex assays.

SARS-CoV-2 infections trigger systemic autoimmune and inflammatory responses which may be regulated through the release of TSLP. Several clinical manifestations of SARS-CoV-2-induced autoimmunity [21,22] and relapse, and/or the exacerbation of other chronic inflammatory diseases have been reported [23,24,25]. TSLP expression contributes to B cell lymphopoiesis and causes an expansion of nearly all immature and mature B cell populations in mice [26], which may lead to the clonal expansion of autoantibody-producing B cells. Additionally, TSLP in allergic inflammation may result in a reduction in B cell tolerance and has been found to promote B lineage-dependent autoimmunity [26]. However, there is currently no evidence that bronchial epithelial-derived TSLP induced by SARS-CoV-2 contributes to the development of autoimmunity or other chronic inflammatory diseases, and these aspects will have to be investigated further.

Elevated plasma TSLP levels in this report are supported by our observations using an immunofluorescence approach in cultured epithelial cells to detect in vitro TSLP expression during SARS-CoV-2 infection. A significant increase in TSLP immunofluorescence was observed upon infection by SARS-CoV-2 in NHBEs, suggesting the de novo synthesis of this cytokine in response to the virus. NHBE cells have previously been used as a model for SARS-CoV-2 studies to examine host immune responses and complement complement pathway activation [4,27]. SARS-CoV-2-induced TSLP expression was confirmed in HBEC samples directly cultured from normal healthy participants. HBECs appeared to have a greater range of TSLP immunofluorescence over that of NHBE, suggesting the differential expression of TSLP within the total population of bronchial samples. The pattern of TSLP immunofluorescence in both mock- and virus-infected cells appeared to be membrane bound with punctate perinuclear patterns. SARS-CoV-2 infection induced elevated immunofluorescence intensity throughout cells and did not appear concentrated in specific regions of the cell. However, further analysis using organelle-specific markers would need to be carried out to determine sites of TSLP trafficking in virus-infected cells.

Interestingly, no significant differences in supernatant TSLP was observed when comparing mock to SARS-CoV-2 infection after 24 h in two other epithelial cell cultures from different tissue sources (nasal and colon tissues). This suggests that the production and release of TSLP from bronchial epithelial cells is not a universal response to SARS-CoV-2 infection amongst epithelial cells in the body and appears to be specific to the lower airway epithelium. It would be interesting to explore TSLP expression in other epithelial cells from other tissue sources beyond this study.

In addition, we found significantly elevated IL-15 in plasma from hospitalized COVID-19 patients at equivalent levels to those previously reported [28]. IL-15 is a pro-inflammatory cytokine that induces the proliferation, survival, and function of lymphoid cell populations, particularly NK cells, and CD8^+^ cytotoxic T cells, as well as epithelial cells [29]. Numerous reports have shown that IL-15 is increased in the plasma of COVID-19 patients, suggesting that IL-15 may be a marker of symptomatic progression or mortality [28,30].

A more substantial 10-fold increase in CXCL-10/IP-10 was observed in hospitalized COVID-19 patients compared with healthy controls. CXCL10/IP-10 is a pro-inflammatory chemokine that attracts a broad range of leukocytes including monocytes/macrophages, B cells, T cells, and other cell types [31]. It has been reported that CXCL10/IP-10 is elevated in COVID-19 patients and may be a predictor of COVID-19 outcome [32]. Moreover, CXCL10/IP-10 in blood has been shown to correlate with increased disease severity in COVID-19 [30,33]. 

A major strength of this study is that our findings demonstrate for the first time that plasma TSLP may be a determinant of the period of hospitalization in COVID-19 patients. The clinical significance of this finding is that SARS-CoV-2-induced TSLP could be a useful indicator of disease severity, which is measured as length of stay in hospital, as well as an important target for therapeutic intervention at early stages of COVID-19 disease. The SARS-CoV-2-induced cytokine storm is a major clinical risk factor for disease severity leading to ARDS-like illness in COVID-19 [5]. The targeted inhibition of TSLP could help prevent the damaging sequelae of COVID-19. In support of this suggestion, a recent report suggested that alarmin cytokines such as TSLP are targets of major therapeutic interest in COVID-19, although few clinical trials are being conducted at present to target these during COVID-19 infection [34].

Despite these important findings, our study had some limitations that should also be addressed. The sample size was necessarily small, as this was designed as a pilot proof of concept study to evaluate the levels of TSLP following SARS-CoV-2 infection. However, a larger sample size would be required to achieve adequate power to confirm our initial findings. Secondly, we were unable to measure TSLP levels in patients immediately after testing positive. Therefore, the temporality of plasma TSLP levels between the initial virus infection and the day of hospitalization could not be determined. Prospective clinical and experimental studies are required to investigate temporal changes in TSLP over the time. Finally, we were unable to distinguish between long-form and short-form TSLP proteins, which are shown to have opposite immune functions [35], due to the unavailability of antibodies specific for the long form of TSLP. Future studies aimed at determining the immune response pattern of long-form and short-form TSLP, as demonstrated in inflammatory diseases such as ulcerative colitis [35], particularly in viral infections, would be of scientific interest.

## 5. Conclusions

In conclusion, our study has generated novel findings regarding the involvement of TSLP in SARS-CoV-2 as a determinant of the period of hospitalization. Our findings also suggest that this is an important inflammatory mechanism and speculate that this mechanism may be amenable to therapeutic intervention in patients with COVID-19.

## Figures and Tables

**Figure 1 viruses-15-00556-f001:**
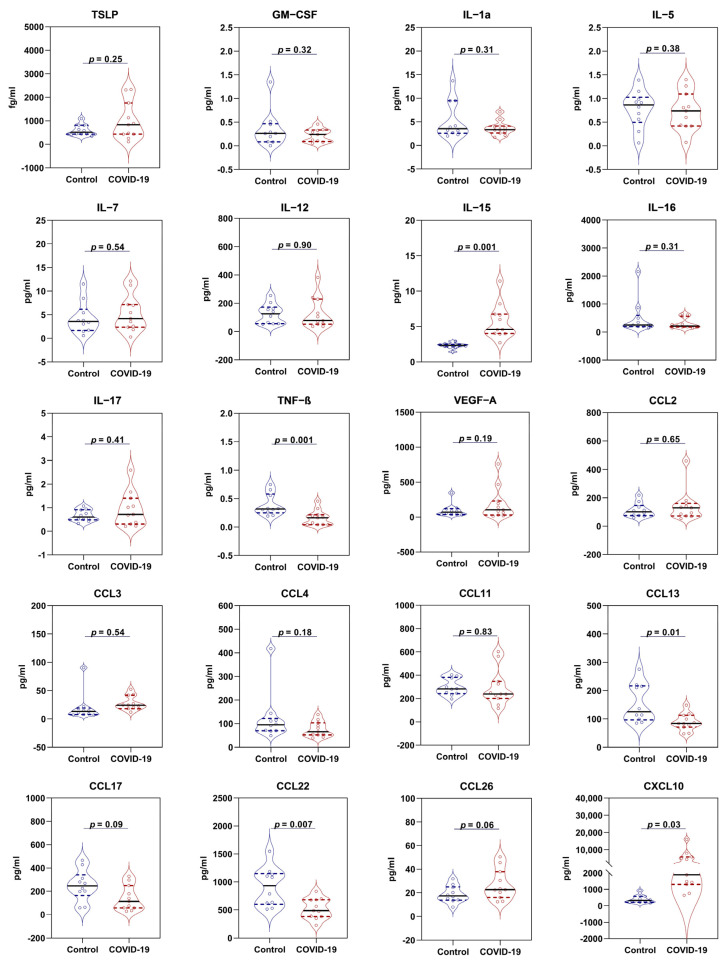
Cytokine profile of study participants. Data presented as mean (solid line, standard error of mean (SEM)), unless otherwise specified. Quartile values presented by dashed lines. *p* values were obtained from one-way analysis of covariance (ANCOVA) adjusted for age and sex. Note that there are missing values (2 measurements) for GM-CSF in the COVID-19 group. For abbreviations, see text.

**Figure 2 viruses-15-00556-f002:**
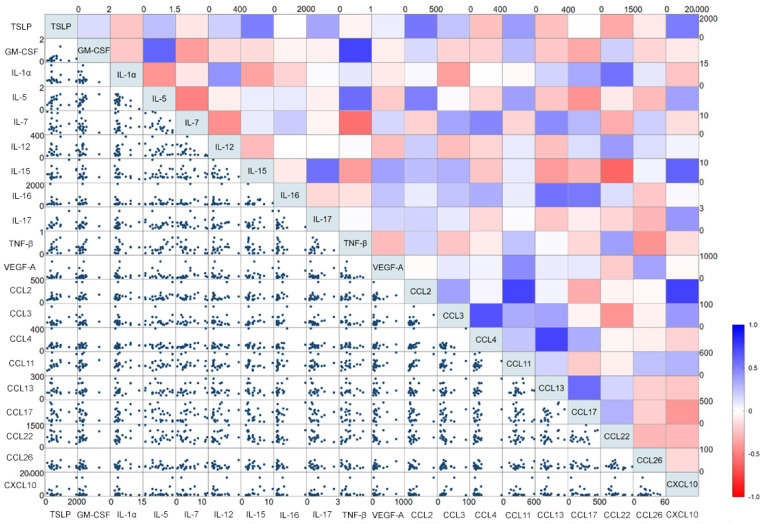
Correlation between plasma TSLP and other cytokines. Bivariate scatterplots of continuous variables are presented below the diagonal; heatmap generated from the Spearman’s correlation coefficients (*ρ*) with corrections for multiple comparisons (Šidák’s correction) are above the diagonal. Units shown are in pg/mL.

**Figure 3 viruses-15-00556-f003:**
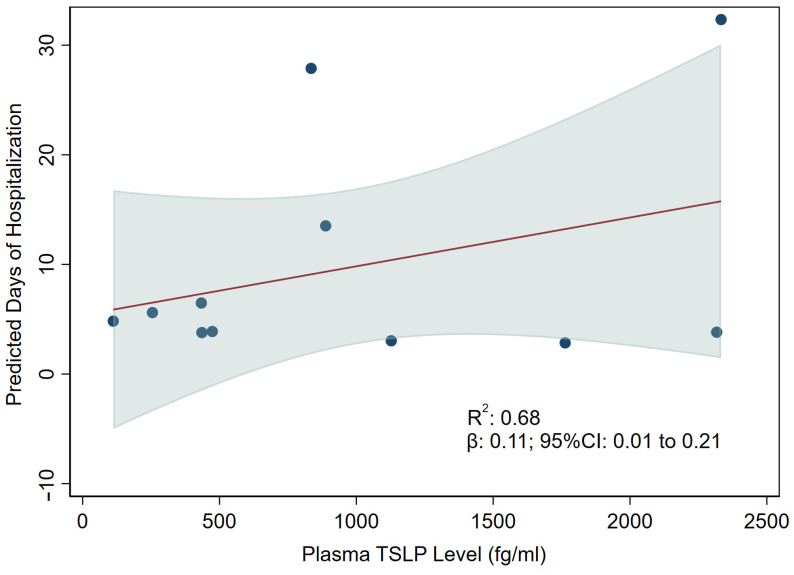
Association between plasma TSLP levels and days of hospitalization in COVID-19 patients. Data presented as regression coefficient (β) (red line) and 95% confidence interval (CI) (shaded area). Multivariable analysis was adjusted for age, sex, heart rate, respiratory rate, body temperature and SpO_2_ on admission, number of comorbidities, and the interval between the date of the COVID-19-positive test result and the date of hospitalization.

**Figure 4 viruses-15-00556-f004:**
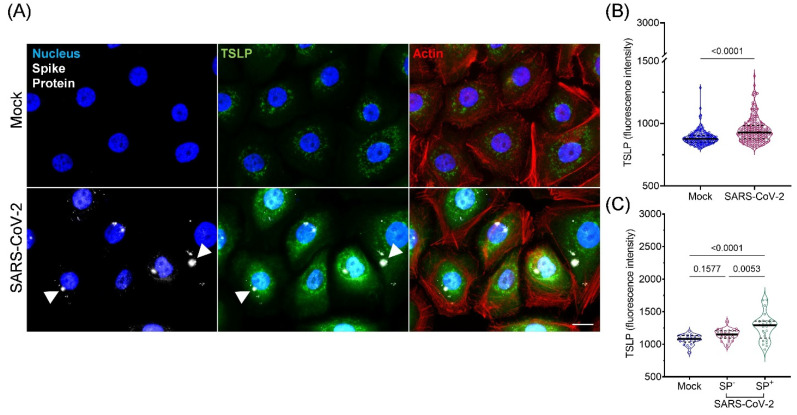
Infection of cultured normal human bronchial epithelial cells (NHBE) with SARS-CoV-2 leads to increased intracellular TSLP. (**A**) NHBE cells were incubated with mock or SARS-CoV-2 (MOI 1) for 24 h. Cells were fixed, permeabilized, and labeled for nucleus (blue), SARS-CoV-2 SP (white), TSLP (green), and actin cytoskeleton (red). Arrowheads indicate punctated regions of SP immunoreactivity. (**B**) Intracellular fluorescent intensity of TSLP in cells, quantified using Volocity software. Each point represents a single cell (*n* = 190 for Mock, *n* = 202 for SARS-CoV-2-infected). (**C**) Fluorescence intensity of TSLP in mock- and SARS-CoV-2-infected cells expressing no detectable levels of SP (SP^−^) compared to cells with detectable SP (SP^+^) in the SARS-CoV-2 population (*n* = 22 for Mock SP^−^, *n* = 27 for SARS-CoV-2 SP^−^, *n* = 25 for SARS-CoV-2 SP^+^). Results shown are representative of 8 separate experiments. Data are presented as individual cells (dots), median (solid line), interquartile range (dotted line) and kernel density (violin). Mann–Whitney and Kruskal–Wallis were used to compare fluorescence in (**B**,**C**), respectively. Scale bar represents 10 μm.

**Figure 5 viruses-15-00556-f005:**
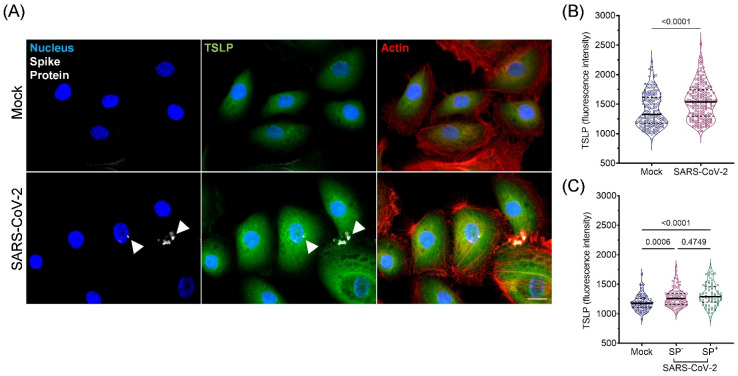
Infection of patient-derived bronchial brushings with SARS-CoV-2. (**A**) Bronchial brushings were collected from an individual donor, grown in culture, and subjected to mock infection (upper panel) or infected with SARS-CoV-2 virus (MOI 1) for 24 h (lower panel). Cells were fixed, permeabilized, and labeled for TSLP (green), SARS-CoV-2 SP (white), nucleus (blue), and actin cytoskeleton (red). Arrowheads indicate SP immunofluorescence in punctate regions. (**B**) Volocity software analysis of intracellular TSLP fluorescence intensity in mock and SARS-CoV-2-infected cells (*n* = 358 for Mock, *n* = 324 for SARS-CoV-2-infected cells). (**C**) Fluorescence intensity of TSLP in mock- and SARS-CoV-2-infected SP^−^ cells compared with SARS-CoV-2 infected SP^+^ cells (*n* = 90 for mock, *n* = 86 for SARS-CoV-2 SP^−^, *n* = 72 for SARS-CoV-2 SP^+^). Results from all donors (*n* = 3) are shown in (**B**) and from a representative donor in (**C**). Data are presented as individual cells (dots), median (solid line), interquartile range (dotted line), and kernel density (violin). Mann–Whitney and Kruskal–Wallis were used to compare fluorescence in (**B**,**C**), respectively. Scale bar represents 10 μm.

## Data Availability

Our data contain confidential personal information and as per provincial regulations, our data cannot be made available to open public database. However, a deidentified dataset with limited variables can be obtained upon reasonable request made to the corresponding author.

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
