# Peer review of "SARS-CoV-2-Induced TSLP Is Associated with Duration of Hospital Stay in COVID-19 Patients"

_viruses, 2023, doi:10.3390/v15020556_

Round 1

Reviewer 1 Report (Previous Reviewer 2)

I think that the manuscript has been improved, and the authors have addressed most of my concerns.

Author Response

Thank you very much for your comment.

Reviewer 2 Report (New Reviewer)

It is of interest that the authors mentioned SARS-CoV-2 induced TSLP is associated with admission length in COVID-19 patients.

However, the authors must discuss the mechanism of the association of SARS-CoV-2 infection with systemic autoimmunity/inflammation, which is close to the development of TSLP.

Quote references below, and re-write the discussion.

1. COVID-19 and autoimmune diseases: is there a connection?

Votto M, Castagnoli R, Marseglia GL, Licari A, Brambilla I.

Curr Opin Allergy Clin Immunol. 2023 Jan 12

2. Coronavirus disease-19 triggered systemic lupus erythematous: A novel entity.

Garg Y, Kakria N, Singh S, Jindal AK.

Lung India. 2023 Jan-Feb;40(1):79-81.

3. [COVID-19 as a trigger of autoimmune hepatitis. Case report].

Volchkova EA, Legkova KS, Topchy TB.

Ter Arkh. 2022 Feb 15;94(2):259-264.

4. Risk for uveitis relapse after COVID-19 vaccination.

Zhong Z, Wu Q, Lai Y, Dai L, Gao Y, Liao W, Feng X, Yang P.

J Autoimmun. 2022 Dec;133:102925. doi: 10.1016/j.jaut.2022.102925. Epub 2022 Oct 4.

5. Molecular mimicry, hyperactive immune system, and SARS-COV-2 are three prerequisites of the autoimmune disease triangle following COVID-19 infection.

Vahabi M, Ghazanfari T, Sepehrnia S.

Int Immunopharmacol. 2022 Nov;112:109183. doi: 10.1016/j.intimp.2022.109183.

6. Lazarian G, Quinquenel A, Bellal M, Siavellis J, Jacquy C, Re D, et al. . Autoimmune Haemolytic Anaemia Associated With COVID-19 Infection. Br J Haematol (2020) 190:29–31. 10.1111/bjh.16794 - DOI - PMC – PubMed

7. Coronavirus Disease 2019 and the Thyroid - Progress and Perspectives.

Inaba H, Aizawa T.Front Endocrinol (Lausanne). 2021 Jun 24;12:708333.

Author Response

Reviewer 2 comment/suggestion

The authors must discuss the mechanism of the association of SARS-CoV-2 infection with systemic autoimmunity/inflammation, which is close to the development of TSLP.

Thank you for your constructive feedback and reference list. We have added a paragraph (line 325) to the Discussion on the potential mechanism of TSLP-directed immune dysregulation during SARS-CoV-2 infection. We hope that this will give better clarity into the association of SARS-CoV-2-induced TSLP with systemic autoimmunity and inflammation.

This manuscript is a resubmission of an earlier submission. The following is a list of the peer review reports and author responses from that submission.

Round 1

Reviewer 1 Report

This study is really very interesting as it shows increased levels of TSLP in the blood of patients with SARS-CoV-2. Besides, it also shows in the cultured human bronchial epithelial cells and patient-derived bronchial epithelial cells - obtained from bronchoscopy- higher TSLP levels after 24 h post-infection with SARS-CoV-2.

I think the article needs the following major revision:

TITLE

The title is too long. It should be more concise and give the reader a key message

ABSTRACT

The Abstract is well written.

INTRODUCTION

The introduction section is adequately written, providing all necessary information to the readers.

Different reports on adults but also in paediatric age showed an increase in IL-6 and TNF-alpha in patients with COVID-19. I suggest adding these references at page 2, Lines 48.

·       “Copaescu, A., Smibert, O., Gibson, A., Phillips, E. J., & Trubiano, J. A. (2020). The role of IL-6 and other mediators in the cytokine storm associated with SARS-CoV-2 infection. The Journal of allergy and clinical immunology, 146(3), 518–534.e1. https://doi.org/10.1016/j.jaci.2020.07.001”

·       “Curatola, A.; Chiaretti, A.; Ferretti, S.; Bersani, G.; Lucchetti, D.; Capossela, L.; Sgambato, A.; Gatto, A. Cytokine Response to SARS-CoV-2 Infection in Children. Viruses 2021, 13, 1868. https://doi.org/10.3390/v13091868”

MATERIAL AND METHODS

In the material and methods section, the study design is well written overall. The description of the statistical analysis performed is also clear and well written

RESULTS

The results of the study are presented concisely and clearly.

DISCUSSION

The discussion is well written, focusing on the key points of the study. As expressed by the authors, the sample is really small, and this necessarily limits the extension of their conclusions to the entire population. I think that on the basis of these data, the authors' statement on TSLP as a therapeutic target for the treatment of COVID-19 is a speculation and I would like to suggest them to formulate this sentence (in the conclusion and in the abstract).

Reviewer 2 Report

  • A brief summary 

In the manuscript, Gerla et al. found some association between plasma TSLP levels and SARS-CoV-2 infection, and further analysis showed that plasma levels of TSLP at the time of hospitalization could plausibly indicate the period of hospitalization in COVID-19 patients. In addition, the authors further found that SARS-CoV-2 virus infection could induce TSLP expression in cultured bronchial epithelial cells (in vitro). Besides, the study also found that plasma levels of IL-15 and CXCL10/IP-10 were elevated in COVID-19 patients over that of normal uninfected controls.

The findings of this study are important in our understanding of the pathogenesis of SARS-CoV-2. The finding will also be helpful in the prevention of SARS-CoV-2, as the study showed that elevated TSLP level upon SARS-CoV-2 infection could be a useful indicator of disease severity.

  • General concept comments

Although in vitro assays showed a significant increase of TSLP in NHBE cells or HBECs upon infection by SARS-CoV-2, the difference in TSLP levels did not achieve a statistically significant level in clinical plasma samples, probably owing to the limited clinical sample size.

While the secretion of TSLP in bronchial epithelial cell cultures was increased upon SARS-CoV-2 infection in vitro, the TSLP in serum can be originated from other cells as well. Recommend testing other cells to validate the findings of the study, such as intestinal or nasal ECs, where the receptor gene of SARS-CoV-2, ACE2 expression level is high, and TSLP is also expressed in these cells.

  • Specific comments:

1)      Abstract section, lines 29-30, “TSLP levels were detected in the plasma of hospitalized COVID-19 patients (603.4 ± 238.5 29 vs 997.6 ± 800.5 fg/ml, mean ± SEM)”, the numbers mentioned here are not matched/consistent with the result section, lines 203-205 “We observed that plasma TSLP levels increased from 603.4 ± 75.4 203 fg/ml in healthy participants (values for plasma samples shown as mean ± SEM) to 997.6 204 ± 241.4 fg/ml in hospitalized COVID-19 patients (Figure 1).”

2)      Line 128, “LLOD of 9.1 fg/ml (LLOQ 34 fg/ml)”, recommends giving the full name of the first appearance abbreviations, for example, “lower limits of detection (LLOD) of 9.1 fg/ml (lower limits of quantitation (LLOQ) 34 fg/ml)”

3)      Line 136, “lower limit of detections (LLODs),” recommends a change to LLODs if the 2) is followed.

4)      Figure 1, line 190, “Missing values for GM-CSF in COVID-19 group (n = 2)”, it appears that there are more than two data points in the COVID-19 group of the subfigure GM-CSF. Besides, in Figure 1, it should be noted what the solid and dash lines in the subfigures stand for in the figure legend.

5)      Line 197 and Line 215, Table 1 is not available either in the main text or supplementary files.

6)      Figure 2, the units of the X-axis and Y-axis should be noted, which correspond with Figure 1.

7)      Line 263, “(p < 0.0001, p = 0.005, respectively, Figure 3C)”, here should be Figure 4C.

8)      Line 253, “Intracellular mean fluorescent intensity of TSLP in cells”. Here, “mean” should be crossed out.

9)      Line 255 and Line 284, same comments as of 8).

10)   Line 285, “TSLP in SP- cells in mock- and SARS-CoV-2-infected cells compared with SP+ cells in SARS-CoV-2 infected cells”, a little confused here, recommend rephrasing to “TSLP in mock-infected and SARS-CoV-2-infected SP- cells compared with SARS-CoV-2 infected SP+ cells”.

11)   Line 286, “(n = 90 for Mock SP-, n = 86 for SARS-CoV-2 SP-, n = 72 for SARS-CoV-2 SP+)”, here, the “SP-” in “n = 90 for Mock SP-” should be crossed out since Mock group was not infected by SARS-CoV-2.
